# Duration of birth interval and its predictors among reproductive-age women in Ethiopia: Gompertz gamma shared frailty modeling

**Getayeneh Antehunegn Tesema** [1]*, **Misganaw Gebrie Worku**[2], **Achamyeleh Birhanu Teshale**[1]

**1** Department of Epidemiology and Biostatistics, Institute of Public Health, College of Medicine and Health Science, University of Gondar, Gondar, Ethiopia, **2** Department Human Anatomy, School of Medicine, of the College of Medicine and Health Science, University of Gondar, Gondar, Ethiopia

* getayenehantehunegn@gmail.com

**Data Availability Statement:** The data used in this study are from the Measure DHS program (www. dhsprogram.com) and can be accessed following the protocol outlined in the Methods section.

## Abstract

### Background

The World Health Organization recommended a minimum of 33 months between consecutive live births to reduce the incidence of adverse pregnancy outcomes. Poorly spaced pregnancies are associated with poor maternal and child health outcomes such as low birth weight, stillbirth, uterine rupture, neonatal mortality, maternal mortality, child malnutrition, and maternal hemorrhage. However, there was limited evidence on the duration of birth interval and its predictors among reproductive-age women in Ethiopia. Therefore, this study aimed to investigate the duration of birth interval and its predictors among reproductive-age women in Ethiopia.

### Methods

A secondary data analysis was conducted based on the 2016 Ethiopian Demographic and Health Survey data. A total weighted sample of 11022 reproductive-age women who gave birth within five years preceding the survey was included for analysis. To identify the predictors, the Gompertz gamma shared frailty model was fitted. The theta value, Akakie Information Criteria (AIC), Bayesian Information Criteria (BIC), and deviance was used for model selection. Variables with a p-value of less than 0.2 in the bi-variable analysis were considered for the multivariable analysis. In the multivariable Gompertz gamma shared frailty analysis, the Adjusted Hazard Ratio (AHR) with a 95% Confidence Interval (CI) was reported to show the strength and statistical significance of the association.

### Results

The median inter-birth interval in Ethiopia was 38 months (95% CI: 37.58, 38.42). Being living in Addis Ababa (AHR = 0.15, 95% CI: 0.03, 0.70), being rural resident (AHR = 1.13, 95% CI: 1.01, 1.23), being Muslim religious follower (AHR = 6.53, 95% CI: 2.35, 18.18), having three birth (AHR = 0.51, 95% CI: 0.10, 0.83), having four birth (AHR = 0.30, 95% CI: 0.09, 0.74), five and above births (AHR = 0.10, 95% CI: 0.02, 0.41), and using contraceptive

**Funding:** The author(s) received no specific funding for this work

**Competing interests:** The authors have declared that no competing interests exist.

**Abbreviations:** AHR, Adjusted Hazard Ratio; AIC, Akaike Information Criteria; BIC, Bayesian Information Criteria; CI, Confidence Interval; CHR, Crude Hazard Ratio; EAs, Enumeration Areas; EDHS, Ethiopian Demographic and Health Survey; IUGR, Intrauterine Growth Restriction; SSA, Sub-Saharan Africa; WHO, World Health Organization.

(AHR = 2.35, 95% CI: 1.16, 4.77) were found significant predictors of duration of birth interval.

## Conclusion

The length of the inter-birth interval was consistent with the World Health Organization recommendation. Therefore, health care interventions that enhance modern contraceptive utilization among women in rural areas and Muslim religious followers would be helpful to optimize birth interval.

## Background

The World Health Organization (WHO) defines the inter-birth interval as the time elapsed between two consecutive live births [1]. The WHO recommends a minimum of 33 months between two successive live births to optimize maternal and newborn health [2, 3]. The short birth interval is defined as the period between birth to successive pregnancy less than 24 months or an inter-birth interval of fewer than 33 months [4]. Like many sub-Saharan African countries, Ethiopia is one of the countries with a high fertility rate (4.3). Which is the second-most populous country in Africa with a population of 114,963,588 [5]. Inter-birth interval has a significant impact on the country's population size and maternal and child health [6, 7].

A short birth interval is responsible for the huge number of maternal and perinatal mortality in low-and middle-income countries [8]. Globally, an estimated 2.6 million stillbirths occur annually [9]. Of which, 67% of the global stillbirth occurred in Sub-Saharan Africa including Ethiopia [10], and most of the stillbirths occur during the intrapartum period that can be preventable by optimizing birth interval [11, 12]. Very short (less than 18–27 months) and very long (typically over 54–59 months) birth intervals are associated with poor health outcomes for both mothers and babies [2]. Previous literature evidenced that both shorter and longer birth intervals have been associated with poor pregnancy outcomes such as low birth weight, Intra-Uterine Growth Restriction (IUGR), prenatal death, Antepartum Hemorrhage (APH), neonatal and maternal mortality [13, 14]. Besides, it imposes a huge financial burden, mental, and psychological consequences to the mothers and the family [4].

Studies on birth interval showed that household wealth status [15], residence [15], husband education [16], maternal education [17, 18], contraceptive use [19, 20], media exposure [21], women health care decision making autonomy [22], religion [23, 24], maternal occupation [15], breastfeeding duration [23], parity [24], and maternal age [25] were significant predictors of birth interval.

Though optimal birth spacing is strongly linked to better health outcomes for both mothers and babies [26], more than half of women in Ethiopia have shorter birth intervals [15]. However, there is limited evidence on the duration of birth interval and its predictors among reproductive-age women in Ethiopia. Therefore, this study aimed to investigate the duration of birth intervals and its predictors among reproductive-age women in Ethiopia. Thus, this study findings would help to work on factors to optimize birth spacing.

## Methods

### Data source and sampling procedure

This research was performed based on data from the 2016 Ethiopian Demographic and Health Survey (EDHS). The 2016 EDHS was the fourth survey conducted every five years in Ethiopia.

It is mainly aimed at producing updated indicators related to health and health. Situated in the Horn of Africa, Ethiopia is the second-most populous country in Africa, next to Nigeria, with a fertility rate of 4.3, it has a total population of 114,963, 588. Nine regional states (Afar, Amhara, Benishangul-Gumuz, Gambella, Harari, Oromia, Somali, Southern Nations, Nationalities and People's Region (SNNPR) and Tigray) and two administrative cities (Addis Ababa and Dire-Dawa) constitute the country. A two-stage cluster sampling technique was employed to select the study participants for EDHS 2016. In the first stage, 645 enumeration areas (EAs) (202 in urban areas) were selected with a probability proportional to the size of the EA, while in the second stage, an average of 28 households was systematically selected for each EA. All reproductive-age women who gave birth within five years preceding the survey in the selected enumeration areas were the study population. A total weighted sample of 11022 women who gave birth within five years preceding the survey was included for this study. The detailed methodological procedure was presented in the full EDHS 2016 report [27].

## Measurement of variables

The birth interval was defined as the duration of months between the birth of the index child and the subsequent live birth. The event was defined as the occurrence of live birth after the index child, while women who did not give birth until the end of the follow-up period were considered as censored. The event was recorded as success "if a woman gave live birth after the index child" or failure "if a woman did not give birth until the end of the follow-up period). The independent variables considered for this study were categorized as socio-demographic and economic variables such as residence, region, religion, maternal education, maternal age, husband education, maternal occupation, sex of household head, media exposure, and wealth status, and maternal obstetric and health services related factors such as parity, women health care decision making autonomy, distance to the health facility, and contraceptive use (Table 1).

## Data management and analysis

The weighted data were used for the overall analysis to take into account the sampling design to get reliable statistical estimates. Descriptive and summary statistics were conducted using STATA version 14 software. The EDHS data violates the independence of observations and equal variance assumptions since women within the same cluster might share similar characteristics than women in another cluster.

As this study considered time to event data, survival analysis was considered. The Kaplan-Meier (K-M) and the log-rank test were done to compare the survival curves across categories of different explanatory variables.

Cox proportional hazard model was fitted to identify the predictors of birth interval. The Proportional Hazard (PH) assumption was assessed using the global Schoenfeld residual test., and the PH assumption was violated (Table 2). Therefore, parametric survival models should be fitted to get a reliable estimate. Because the EDHS data structure was hierarchical, we have checked whether there is clustering or not by running the frailty model (random effect survival model) and the theta was significant at the null model ($\theta = 0.24$, 95% CI: 0.19, 0.30) (LR test of theta = 0: $X^2 = 226.36$, $p < 0.001$) it indicates that there was unobserved heterogeneity or shared frailty, that means women in one cluster were more likely to be correlated within the same cluster.

Shared frailty model with baseline distributions (Weibull, Gompertz, Exponential, log-logistic, and lognormal) and frailty distributions (gamma and inverse Gaussian) were modeled by taking EAs/clusters as a random effect. Gompertz gamma shared frailty model was the best-

**Table 1. List of variables used for analysis and their definition and measurement based on the 2016 EDHS.**

| Variable name | Measurement |
|---|---|
| Birth interval | Duration of months between two successive live births |
| Residence | Urban/rural |
| Region | Region where the mother residing recorded as Tigray, Afar, Amhara, Oromia, Somali, Benishangul, south nation nationalities and peoples, Gambella, Harari, Addis Ababa and Dire Dawa |
| Religion | Maternal religion was categorized as Orthodox, Muslim, Protestant and Others (catholic and traditional) |
| Maternal education | Was categorized as no, primary, and secondary and above |
| Maternal age | Maternal age in year was categorized as 15–19, 20–24, 25–29, 30–34, 35–39, 40–44 and 45–49 |
| Husband education | Education status of their husband was categorized as no, primary, secondary and above |
| Maternal occupation | The occupation status of woman was recorded as working/not working |
| Sex of household | Women were asked about the who is the head of household and answered as male/female |
| Media exposure | Media exposure was calculated by aggregating TV watching, radio listening, and reading newspapers and woman who has exposure to either of the media sources was categorized as having media exposure and the rest considered as having no media exposure. |
| Household wealth status | It was computed based on principal component analysis using the household assets recorded in EDHS and categorized as poorest, poorer, middle, richer and richest |
| Parity | The number of births given before the survey, including the most recent births among women who give birth within five years before the survey. Recoded as ≤2, 3, 4 and ≥5 based on previous literatures. |
| Women health care decision making autonomy | In EDHS 2016 the question was asked as "person who usually decides on the respondent's health care?". The response for this question was respondent alone coded as "1", jointly with their partner coded as "2", and partner alone coded as "3". |
| Distance to health facility | Women perceived distance to health facility and responded as not a big problem/a big problem |
| Contraceptive use | Women asked as whether they used modern contraceptive or not and answered as no/yes |

fitted model since it had the highest values of log-likelihood and AIC. Variable with a p-value less than 0.20 in the bivariable Gompertz gamma shared frailty analysis was included in the multivariable analysis. In the multivariable Gompertz gamma shared frailty analysis, the Adjusted Hazard Ratio (AHR) with 95% Confidence Interval (CI) were reported to declare the strength and significance of the association between birth interval and independent variables.

## Result

### Socio-demographic and economic characteristics of the respondent

A total of 11022 reproductive-age women were included in the study. Of these, 4851 (44.0%) were from the Oromia region and 9807 (89.0%) were rural residents. Nearly one-third (30.4%) of women were aged 25–29 years. The majority (66.1%) of the mothers did not have formal education whereas 37.3% of their husbands attained a primary level of education (Table 3).

### Maternal obstetric and maternal health services related characteristics

Of the total, 4752 (43.1%) of the mothers had five and above births. About 6676 (60.6%) were perceived distance to a health facility as a big problem. More than half (62.3%) of the mothers

**Table 2. Schoenfeld residual test for checking proportional hazard assumption for the duration of birth interval and its predictors among reproductive age women in Ethiopia, 2016.**

| Variables | Rho | Chi2 | Df | Prob>chi2 |
|---|---|---|---|---|
| Region | 0.028 | 2.73 | 1 | 0.09 |
| Residence | -0.070 | 19.33 | 1 | 0.0001 |
| Sex of household head | -0.0008 | 0.001 | 1 | 0.96 |
| Wealth status | 0.034 | 3.72 | 1 | 0.054 |
| Maternal education | -0.057 | 11.95 | 1 | 0.0005 |
| Parity | 0.16 | 59.73 | 1 | 0.001 |
| Husband education | 0.011 | 0.421 | 1 | 0.52 |
| Religion | -0.024 | 1.69 | 1 | 0.19 |
| Contraceptive use | -0.0001 | 0.001 | 1 | 0.99 |
| Women autonomy | 0.014 | 0.62 | 1 | 0.43 |
| Parity | 0.16 | 59.73 | 1 | 0.0001 |
| Husband occupation | 0.012 | 0.49 | 1 | 0.48 |
| Maternal occupation | -0.046 | 7.61 | 1 | 0.006 |
| Health care access problem | -0.0009 | 0.00 | 1 | 0.96 |
| Global test | | 115.94 | 13 | 0.00001 |

made health care decisions jointly with her husband and 31.3% of the women using one of the modern contraceptives (Table 4).

## Predictors of birth interval

The median duration of birth interval in Ethiopia was 38 months (95% CI: 37.58, 38.42). The log-rank test found that residence, maternal education, husband education, maternal occupation, husband occupation, religion, parity, contraceptive use, health care access problem, health care decision making autonomy, wealth status, and sex of household head showed a statistically significant difference in probability of giving birth after the index child (log-rank, p<005) (Table 5). Based on deviance, AIC, BIC, and theta value, the shared frailty with Gompertz distribution and gamma frailty was the best-fitted model for the data (Table 6).

In the Gompertz gamma shared frailty model; religion, residence, contraceptive use, region, and parity preceding birth interval and birth size were significant predictors of birth interval. Women in Addis Ababa were 85% less likely (AHR = 0.15, 95% CI: 0.03, 0.70) to have subsequent birth compared to women in the Oromia region. Rural resident women were 1.13 times (AHR = 1.13, 95% CI: 1.01, 1,23) more likely to give subsequent birth than urban residents. Muslim and other (protestant and catholic) religious follower women were 6.53 times (AHR = 6.53, 95% CI: 2.35, 18.18) and 4.03 times (AHR = 4.03, 95% CI: 1.40, 11.60) more likely to have subsequent birth compared to orthodox Christian followers, respectively. Women who had three births, fourth births, and five and above births were 49% (AHR = 0.51, 95% CI: 0.10, 0.83), 70% (AHR = 0.30, 95% CI: 0.09, 0.74), and 90% (AHR = 0.10, 95% CI: 0.02, 0.41) less likely to have subsequent birth than women who had two and fewer than two births, respectively. Besides, women who did not use any kind of contraceptive were 2.35 times (AHR = 2.35, 95% CI: 1.16, 4.77) more likely to have subsequent birth compared to contraceptive users (Table 6).

## Discussion

In this study, the median birth interval among reproductive age women in Ethiopia was 38 months (95% CI: 37.58, 38.42). This finding was higher than the study findings [28, 29] and

**Table 3. Socio-demographic characteristics of reproductive-age women who gave birth in the last five years in Ethiopia, 2016.**

| Variables | Weighted frequency (n = 11022) | Percentage (%) |
|---|---|---|
| **Region** | | |
| Tigray | 716 | 6.5 |
| Afar | 114 | 1.0 |
| Amhara | 2072 | 18.8 |
| Oromia | 4851 | 44.0 |
| Somali | 508 | 4.6 |
| Benishangul Gumuz | 122 | 1.1 |
| SNNPRs | 2296 | 20.8 |
| Gambella | 27 | 0.2 |
| Harari | 26 | 0.2 |
| Addis Ababa | 244 | 2.2 |
| Dire Dawa | 47 | 0.4 |
| **Residence** | | |
| Urban | 1215 | 11.0 |
| Rural | 9807 | 89.0 |
| **Maternal age** | | |
| 15–19 | 378 | 3.4 |
| 20–24 | 2068 | 18.8 |
| 25–29 | 3353 | 30.4 |
| 30–34 | 2489 | 22.6 |
| 35–39 | 1772 | 16.1 |
| 40–44 | 723 | 6.6 |
| 45–49 | 239 | 2.2 |
| **Maternal education** | | |
| No | 7284 | 66.1 |
| Primary | 2950 | 26.8 |
| Secondary and above | 788 | 7.1 |
| **Husband education** | | |
| No | 5003 | 45.4 |
| Primary | 4115 | 37.3 |
| Secondary and above | 1904 | 17.3 |
| **Religion** | | |
| Orthodox | 3772 | 34.2 |
| Muslim | 4561 | 41.4 |
| Protestant | 2329 | 21.1 |
| Others | 360 | 3.3 |
| **Wealth status** | | |
| Poorest | 2636 | 23.9 |
| Poorer | 2520 | 22.9 |
| Middle | 2280 | 20.7 |
| Rich | 1998 | 18.1 |
| Richest | 1588 | 14.4 |
| **Sex of household head** | | |
| Male | 9493 | 86.1 |
| Female | 1529 | 13.9 |
| **Media exposure** | | |

(*Continued*)

**Table 3.** (Continued)

| Variables | Weighted frequency (n = 11022) | Percentage (%) |
|---|---|---|
| No | 7375 | 66.9 |
| Yes | 3647 | 33.1 |
| **Maternal occupation** | | |
| Farmer | 2459 | 22.3 |
| Government employee | 159 | 1.4 |
| Merchant | 1655 | 15.0 |
| Others | 6749 | 1.2 |
| **Husband occupation** | | |
| Farmer | 6887 | 62.5 |
| Government employee | 416 | 3.8 |
| Merchant | 1019 | 9.3 |
| Others | 2699 | 24.5 |

WHO recommendations [30]. This could be due to the establishment of health extension workers and the expansion of primary health care units that could increase access to family planning services in Ethiopia [31]. Family planning service utilization such as modern contraceptive use is identified as the key intervention strategy to optimize birth spacing, contraceptive users have long birth interval than non-users. Besides, currently in Ethiopia, several public health programs are working on enhancing women empowerment and maternal education, this could contribute to this difference.

In the Gompertz gamma shared frailty model; residence, region, religion, parity, and contraceptive use were the significant predictors of birth interval. Women living in Addis Ababa were less likely to have subsequent birth as compared to women living in the Oromia region. This could be due to the difference in availability and accessibility of maternal health services such as family planning services and access to health information [32]. In Addis Ababa, women may have a better socioeconomic status and education resulting in better health care

**Table 4. Maternal obstetrical and health services related characteristics of the respondents, 2016.**

| Variable | Weighted frequency | Percentage (%) |
|---|---|---|
| **Parity** | | |
| ≤ 2 | 3181 | 28.9 |
| 3 | 1655 | 15.0 |
| 4 | 1434 | 13.0 |
| ≥ 5 | 4752 | 43.1 |
| **Distance to health facility** | | |
| Not a big problem | 4346 | 39.4 |
| Big problem | 6676 | 60.6 |
| **Health care decision making autonomy** | | |
| Respondent alone | 1362 | 12.4 |
| Jointly with husband/partner | 6869 | 62.3 |
| Husband/respondent alone | 2791 | 25.3 |
| **Contraceptive use** | | |
| Modern contraceptive | 3449 | 31.3 |
| Traditional | 47 | 0.4 |
| Not using | 7526 | 68.3 |

**Table 5. Log rank test for the predictors of birth interval.**

| Variable | p-value | Variable | p-value |
|---|---|---|---|
| Husband education | 0.001 | Parity | 0.005 |
| Residence | 0.0001 | Maternal occupation | 0.001 |
| Sex of household head | 0.0022 | Husband occupation | 0.006 |
| Wealth index | 0.0001 | Contraceptive use | <0.001 |
| Distance to health facility | <0.001 | Religion | 0.001 |
| Maternal education | 0.001 | Women autonomy | 0.041 |

knowledge towards reproductive health care services such as family planning use [33]. Besides, women in Addis Ababa are more educated and therefore they understand the health implications of closely spaced birth intervals, this could enhance women to have optimal birth intervals [34].

Women who were Muslim and other (catholic and protestant) religious followers were more likely to have subsequent birth than Orthodox Christian religious followers. It is consistent with studies reported in Netherland [35], Saudi Arabia [36], and Manipur [37], it is mainly related to the difference in community practices across regions [38, 39]. In the Ethiopian context, family planning methods utilization such as modern contraceptive use is poor in Muslims and Protestants religious followers this might be the possible reason for the short interbirth intervals [40]. Besides, evidence suggested that Muslims' attitude towards family planning use is unfavorable and have a lower approval rate towards contraceptive use [41].

Parity was a significant predictor of birth interval. Women who had three births and above were less likely to have subsequent birth than women who had two birth and less. This is consistent with study findings in Tanzania [6], Bangladesh [4], and Iran [17]. The possible justification may be because mothers who have more than two births may not desire extra children since they face an economic burden to raise their children [42]. Besides, multiparous women are at increased risk of pregnancy-related complications they need to have adequate birth spacing to restore from the previous pregnancy and lactation to get a good pregnancy outcome [43].

Women who were not using contraceptives had a higher hazard of giving subsequent birth compared to women who used contraceptives. It is in line with study findings [15, 19, 44], this could be since contraceptive use is one of the most important factors affecting birth interval, and it's identified as the key strategy to optimize birth interval [19, 45]. Women who use modern contraceptive space births longer than non-users [20]. The hazard of having subsequent birth among rural residents was higher than women living in urban areas. It is consistent with previous studies [46, 47], the possible explanation might be due to the residential variation in reproductive health care services. Urban women have good access to family planning services than rural residents as many of the health facilities are highly concentrated in urban areas. Besides, urban women are more educated and know the maternal and child health implication of short birth intervals [48].

The result of this study should be interpreted in light of the following limitations. First, the EDHS survey did not incorporate clinically confirmed data rather it relied on mothers' or caregivers' reports and might have the possibility of social desirability and recall bias. These might underestimate, overestimate, or reverse the strength of association. Besides, the cross-sectional nature of the study does not allow to infer the temporal relationship between birth interval and the factors. Despite the abovementioned limitations, this study has numerous strengths. First, the study is based on weighted EDHS data that can be generalizable to reproductive-age

**Table 6. The bi-variable and multivariable Gompertz gamma shared frailty model for the predictors of birth interval in Ethiopia, 2016.**

| Variables | Birth status | | Hazard Ratio (HR) with 95% CI | |
|---|---|---|---|---|
| | Event | Censored | CHR with 95%% CI | AHR 95 CI% |
| **Region** | | | | |
| Oromia | 1673 | 3177 | 1 | 1 |
| Tigray | 170 | 546 | 0.27 (0.08, 0.93) | 1.47 (0.35, 6.19) |
| Afar | 37 | 72 | 0.34 (0.09, 1.30) | 0.37 (0.09, 1.55) |
| Amhara | 413 | 1659 | 0.06 (0.02, 0.21) | 0.58 (0.16, 2.08) |
| Somali | 234 | 273 | 0.70 (0.21, 2.34) | 0.83 (0.23, 2.93) |
| Benishangul-Gumuz | 40 | 81 | 0.15 (0.04, 0.57) | 0.37 (0.10, 1.30) |
| SNNPRs | 672 | 1624 | 0.17 (0.05, 0.57) | 0.46 (0.14, 1.45) |
| Gambella | 6 | 21 | 0.61 (0.17, 2.12) | 3.10 (0.79, 12.25) |
| Harari | 8 | 18 | 0.08 (0.02, 0.36) | 0.32 (0.08, 1.25) |
| Addis Ababa | 41 | 202 | 0.01 (0.002, 0.05) | 0.15 (0.03, 0.70)* |
| Dire Dawa | 13 | 33 | 0.30 (0.07, 1.18) | 0.42 (0.10, 1.74) |
| **Residence** | | | | |
| Urban | 227 | 988 | 1 | 1 |
| Rural | 3086 | 6721 | 1.32 (1.16, 1.52) | 1.13 (1.01, 1.23)* |
| **Maternal education status** | | | | |
| No | 2420 | 4864 | 1 | 1 |
| Primary | 768 | 2182 | 1.59 (1.44, 1.74) | 1.50 (0.74, 3.04) |
| Secondary and higher | 125 | 663 | 1.19 (1.01, 1.42) | 0.52 (0.19, 1.47) |
| **Husband education status** | | | | |
| No | 1602 | 3401 | 1 | 1 |
| Primary | 1347 | 2768 | 1.41 (1.29, 1.54) | 1.03 (0.46, 2.32) |
| Secondary and higher | 364 | 1540 | 1.05 (0.94, 1.16) | 0.56 (0.23, 1.37) |
| **Wealth status** | | | | |
| Poorest | 965 | 1671 | 1 | 1 |
| Poorer | 844 | 1675 | 0.88 (0.79, 0.99) | 1.44 (0.57, 3.64) |
| Middle | 668 | 1612 | 0.78 (0.69, 0.88) | 0.85 (0.32, 2.26) |
| Richer | 534 | 1464 | 0.73 (0.64, 0.83) | 0.77 (0.23, 2.56) |
| Richest | 302 | 1286 | 0.63 (0.55, 0.72) | 0.69 (0.17, 2.91) |
| **Sex of household head** | | | | |
| Male | 2917 | 6576 | 1 | 1 |
| Female | 395 | 1534 | 1.66 (0.83, 3.35) | 1.41 (0.69, 2.88) |
| **Religion** | | | | |
| Orthodox | 829 | 2942 | 1 | 1 |
| Muslim | 1701 | 2860 | 2.18 (1.96, 2.42) | 6.53 (2.35, 18.18)* |
| Others* | 783 | 1907 | 1.46 (1.28, 1.67) | 4.03 (1.40, 11.60)* |
| **Maternal occupation** | | | | |
| Agricultural employee | 461 | 1194 | 1 | 1 |
| Government employee | 23 | 136 | 1.11 (0.80, 1.53) | 0.54 (0.13, 3.32) |
| Merchant | 686 | 1772 | 0.99 (0.85, 1.14) | 2.29 (0.81, 6.45) |
| Others | 2143 | 4606 | 1.37 (1.22, 1.54) | 1.54 (0.67, 3.53) |
| **Husband occupation** | | | | |
| Agricultural employee | 278 | 741 | 1 | 1 |
| Government employee | 102 | 314 | 1.05 (0.88, 1.25) | 0.54 (0.13, 2.32) |
| Merchant | 2208 | 4679 | 0.91 (0.80, 1.03) | 2.29 (0.81, 6.45) |
| Other | 724 | 1975 | 0.77 (0.68, 0.88) | 2.07 (0.79, 5.47) |

(*Continued*)

**Table 6.** (Continued)

| Variables | Birth status | | Hazard Ratio (HR) with 95% CI | |
|---|---|---|---|---|
| | Event | Censored | CHR with 95%% CI | AHR 95 CI% |
| **Parity** | | | | |
| ≤ 2 | 543 | 2638 | 1 | 1 |
| 3 | 542 | 1113 | 0.98 (0.86, 1.11) | 0.51 (0.10, 0.83)* |
| 4 | 508 | 925 | 0.68 (0.60, 0.77) | 0.30 (0.09, 0.74)* |
| > 4 | 1719 | 3033 | 0.22 (0.20, 0.24) | 0.10 (0.02, 0.41)* |
| **Women health care decision making autonomy** | | | | |
| Respondent alone | 359 | 1003 | 1 | 1 |
| Jointly with husband/partner | 2082 | 4788 | 1.22 (1.10, 1.35) | 1.92 (0.89, 4.14) |
| Husband/parent only | 873 | 1918 | 0.99 (0.99, 1.25) | 1.94 (0.74, 5.10) |
| **Contraceptive use** | | | | |
| Yes | 750 | 2746 | 1 | 1 |
| No | 2563 | 4963 | 1.39 (1.25, 1.53) | 2.35 (1.16, 4.77)* |

AHR: Adjusted Hazard Ratio, CHR: Crude Hazard Ratio, CI: Confidence Interval.

women in Ethiopia. Moreover, the use of advanced statistical modeling that took into account the nested nature of the DHS data to get reliable standard error and estimate.

## Conclusion

Though reproductive health services such as family planning services have expanded, inadequate birth interval remains a major public health care concern in Ethiopia. Place of residence, religion, parity, contraceptive use, and the region were found significant predictors of birth interval. These findings highlight that health programs working on improving maternal health care access such as family planning services should be scaled up to optimize birth interval in rural residents. Besides, health care programs should work in collaboration with religious leaders about modern contraceptive utilization to have adequate birth interval to improve maternal and child health outcomes.

## Acknowledgments

We would like to thank the measure DHS program for providing the data set.

## Author Contributions

**Conceptualization:** Getayeneh Antehunegn Tesema, Misganaw Gebrie Worku, Achamyeleh Birhanu Teshale.

**Data curation:** Getayeneh Antehunegn Tesema, Achamyeleh Birhanu Teshale.

**Formal analysis:** Getayeneh Antehunegn Tesema, Misganaw Gebrie Worku, Achamyeleh Birhanu Teshale.

**Investigation:** Getayeneh Antehunegn Tesema, Misganaw Gebrie Worku, Achamyeleh Birhanu Teshale.

**Methodology:** Getayeneh Antehunegn Tesema, Misganaw Gebrie Worku, Achamyeleh Birhanu Teshale.

**Software:** Getayeneh Antehunegn Tesema, Misganaw Gebrie Worku, Achamyeleh Birhanu Teshale.

**Supervision:** Misganaw Gebrie Worku.

**Visualization:** Getayeneh Antehunegn Tesema, Misganaw Gebrie Worku, Achamyeleh Birhanu Teshale.

**Writing – original draft:** Getayeneh Antehunegn Tesema, Misganaw Gebrie Worku, Achamyeleh Birhanu Teshale.

**Writing – review & editing:** Getayeneh Antehunegn Tesema, Misganaw Gebrie Worku, Achamyeleh Birhanu Teshale.

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
