## [Decision Letter · Decision Letter 0]

1 Dec 2020

PONE-D-20-31922

Duration of birth interval and its predictors among reproductive-age women in Ethiopia: Gompertz Gamma Shared Frailty Modeling

PLOS ONE

Dear Dr. Tesema,

Thank you for submitting your manuscript to PLOS ONE. After careful consideration, we feel that it has merit but does not fully meet PLOS ONE’s publication criteria as it currently stands. Therefore, we invite you to submit a revised version of the manuscript that addresses the points raised during the review process.

Two experts in the field handled your manuscript, and we are very thankful for their time and contributions. Although some interest was found in your study, several major concerns arose that overshadowed the enthusiasm for your results. Please address ALL of the reviewers' comments in your revised manuscript. In addition, you must contact a copyeditor of your choosing to proof your revised manuscript for grammar and standard English.

We look forward to receiving your revised manuscript.

Kind regards,

Frank T. Spradley

Academic Editor

PLOS ONE

2.In your Data Availability statement, you have not specified where the minimal data set underlying the results described in your manuscript can be found. PLOS defines a study's minimal data set as the underlying data used to reach the conclusions drawn in the manuscript and any additional data required to replicate the reported study findings in their entirety. All PLOS journals require that the minimal data set be made fully available. For more information about our data policy, please see http://journals.plos.org/plosone/s/data-availability.

Reviewers' comments:

Reviewer's Responses to Questions

**Comments to the Author**

1. Is the manuscript technically sound, and do the data support the conclusions?

Reviewer #1: Yes

Reviewer #2: Yes

2. Has the statistical analysis been performed appropriately and rigorously? 

Reviewer #1: Yes

Reviewer #2: Yes

3. Have the authors made all data underlying the findings in their manuscript fully available?

Reviewer #1: Yes

Reviewer #2: Yes

4. Is the manuscript presented in an intelligible fashion and written in standard English?

Reviewer #1: No

Reviewer #2: Yes

5. Review Comments to the Author

Reviewer #1: -The study topic is important, however the manuscript is mainly prepared as a report not an original paper.

-The abstract provides well-known facts, and the conclusion is a repetition of the results.

-The introduction should be summarized.

-More information should be provided about geographical and socio-demographic factors of the area under study, especially for comparison of data related to different cities.

-The discussion section should begin with the information and message(s) of the current study, but it is begun by items presented in the introduction section.

-The interpretation of findings should be expanded in the discussion section.

-The study limitations should be expanded.

-The conclusion is too vague.

-The English writing should be improved.

Reviewer #2: - The study aimed to investigate the birth interval and its predictors among reproductive age women in Ethiopia.

- The authors used EDHS-2016 online accessible dataset and applied statistical tests for survival analysis by Gompertz gamma shared frailty modeling.

- However, the predictors determined by the authors as maternal education, parity, economic status, contraceptive use are already well known factors responsible for lesser birth intervals (Hailu and Gulte, 2016; Molitoris, 2018). Authors may emphasize their findings with respect to existing literature (in other populations/Ethiopia population if any). In addition, the quality of the manuscript would be improved if the authors can include or use these predictors to predict the birth interval and make it available online as a tool.

- The manuscript can be re-checked for grammatical errors.

6. PLOS authors have the option to publish the peer review history of their article (what does this mean?). If published, this will include your full peer review and any attached files.

Reviewer #1: **Yes: **Roya Kelishadi

Reviewer #2: No

---

## [Author Response · Author response to Decision Letter 0]

21 Dec 2020

Point by point response for editors/reviewers comments 

Manuscript title: Duration of birth interval and its predictors among reproductive-age women in Ethiopia: Gompertz Gamma Shared Frailty Modeling

Manuscript ID: PONE-D-20-31922

Dear editor/reviewer. 

Dear all,

We would like to thank you for this constructive, building, and improvable comments on this manuscript that would improve the substance and content of the manuscript. We considered each comment and clarification questions of editors and reviewers on the manuscript thoroughly. Our point-by-point responses for each comment and question are described in detail on the following pages. Further, the details of changes were shown by track changes in the supplementary document attached.

Editors comment

1. Although some interest was found in your study, several major concerns arose that overshadowed the enthusiasm for your results. Please address ALL of the reviewers' comments in your revised manuscript. Besides, you must contact a copyeditor of your choosing to proof your revised manuscript for grammar and standard English.

Authors’ response: Thank you editor for the comments. We extensively modified the sentence structure, wording, spelling, and punctuations with the help of language experts at the university. Besides, we address all the comments raised by the reviewers.

Response to reviewer’s comment

Reviewer#1

1. The study topic is important; however, the manuscript is mainly prepared as a report not an original paper.

-The abstract provides well-known facts, and the conclusion is a repetition of the results.

Authors’ response: Thank you reviewers for the comments. We used the EDHS data for this study to investigate the duration of birth interval and its predictors, as the EDHS reported only the descriptive statistics and did not address what are the factors associated with birth interval duration. Besides, the median duration of the birth interval was not reported using advanced statistical models such as frailty models to take into account the hierarchical nature of EDHS data to draw a valid conclusion. Therefore, we have done this study to estimate the median duration of birth interval and identify significant predictors that determine the length of the birth interval. We extensively rewrite the abstract of the manuscript including the conclusion section. (See the revised manuscript, Abstract section, line 12-40, page 2-3)

2. The introduction should be summarized.

Authors’ response: Thank you editor for the constructive comment. We extensively organize literature and summarized the introduction section. (See the revised manuscript, line 41-71, page 4-5)

3. More information should be provided about geographical and socio-demographic factors of the area under study, especially for comparison of data related to different cities.

Authors’ response: Thank you reviewer for the comments. As per your critical recommendation, we incorporated the geographical and socio-demographic characteristics of the country such as fertility rate, total population size, and the number of regions as these variables are linked with our outcome variables "birth interval". (See the revised manuscript, line 73-81, page 5)

4. The discussion section should begin with the information and message(s) of the current study, but it is begun by items presented in the introduction section. The interpretation of findings should be expanded in the discussion section.

Authors’ response: Thank you for the comments. We begin the discussion section with the current findings of the study and the interpretation of the findings in the discussion section is expanded with appropriate citations. (See the revised manuscript, Discussion section, line 172-227, page 9-12)

5. The study limitations should be expanded. The conclusion is too vague.

Authors’ response: Thank you reviewer for the comments. We accept the comments and modified the limitation and conclusion section of the study. (See the revised manuscript, Line 219-236, page 11-12)

6. English writing should be improved.

Authors’ response: Thank you reviewer for the comments. We extensively edited the whole document for any grammatical error and sentence structure with the help of language experts in the university. (See the revised manuscript)

Reviewer #2

1. The study aimed to investigate the birth interval and its predictors among reproductive age women in Ethiopia. The authors used EDHS-2016 online accessible dataset and applied statistical tests for survival analysis by Gompertz gamma shared frailty modeling. However, the predictors determined by the authors as maternal education, parity, economic status, contraceptive use are already well known factors responsible for lesser birth intervals (Hailu and Gulte, 2016; Molitoris, 2018). Authors may emphasize their findings with respect to existing literature (in other populations/Ethiopia population if any). In addition, the quality of the manuscript would be improved if the authors can include or use these predictors to predict the birth interval and make it available online as a tool.

Authors’ response: Thank you reviewer for the comments. As you stated very well most of the variables are reported by previous researchers but our study is based on the weighted nationally representative data using advanced statistical models such as Gompertez gamma shared frailty modeling to make valid inferences. Besides, big data were used in this study and therefore, the study has a high power to detect the true effect of the variables. We have incorporated the tool/independent variables we used in the study in the form of a table. (See Table 1)

2. - The manuscript can be re-checked for grammatical errors.

Authors’ response: Thank you reviewer for the comments. We extensively modified the manuscript for editorial as well as typographical errors. (See the revised manuscript)

---

## [Decision Letter · Decision Letter 1]

2 Feb 2021

Duration of birth interval and its predictors among reproductive-age women in Ethiopia: Gompertz Gamma Shared Frailty Modeling

PONE-D-20-31922R1

Dear Dr. Tesema,

We’re pleased to inform you that your manuscript has been judged scientifically suitable for publication and will be formally accepted for publication once it meets all outstanding technical requirements.

Kind regards,

Frank T. Spradley

Academic Editor

PLOS ONE

Additional Editor Comments (optional):

Reviewers' comments:

Reviewer's Responses to Questions

**Comments to the Author**

1. If the authors have adequately addressed your comments raised in a previous round of review and you feel that this manuscript is now acceptable for publication, you may indicate that here to bypass the “Comments to the Author” section, enter your conflict of interest statement in the “Confidential to Editor” section, and submit your "Accept" recommendation.

Reviewer #1: All comments have been addressed

2. Is the manuscript technically sound, and do the data support the conclusions?

Reviewer #1: Yes

3. Has the statistical analysis been performed appropriately and rigorously? 

Reviewer #1: Yes

4. Have the authors made all data underlying the findings in their manuscript fully available?

Reviewer #1: Yes

5. Is the manuscript presented in an intelligible fashion and written in standard English?

Reviewer #1: Yes

6. Review Comments to the Author

Reviewer #1: The revised paper has improved in terms of scientific and structural aspects. Authors have succeeded to make necessary revision.

7. PLOS authors have the option to publish the peer review history of their article (what does this mean?). If published, this will include your full peer review and any attached files.

Reviewer #1: **Yes: **Roya Kelishadi

---

## [Editor Report · Acceptance letter]

8 Feb 2021

PONE-D-20-31922R1 

Duration of birth interval and its predictors among reproductive-age women in Ethiopia: Gompertz Gamma Shared Frailty Modeling 

Dear Dr. Tesema:

I'm pleased to inform you that your manuscript has been deemed suitable for publication in PLOS ONE. Congratulations! Your manuscript is now with our production department. 

Kind regards, 

on behalf of

Dr. Frank T. Spradley 

Academic Editor

PLOS ONE